# Inducing Causal Order through Tabular In-Context Learning

**Sascha Xu** [* 1]  **Sarah Mameche** [* 1]  **Jilles Vreeken** [1]

## Abstract

Tabular foundation models (TFMs) achieve state-of-the-art predictive performance on tabular data, but primarily rely on correlational structure susceptible to change under distribution shift or intervention. To address this, we propose inducing causal structure into TFMs by constraining predictive attention to features that precede a given target under a causal ordering. Our architecture differentiably ranks tabular columns and constructs an attention mask that restricts information flow in the anticausal direction. Crucially, we infer the causal order itself directly from data by likelihood maximization without access to the underlying causal structure during training. Experiments across diverse classes of structural causal models show that our method TABORDER reliably infers accurate causal orderings, enabling it to outperform purely associational models under interventions and distribution shift.

## 1. Introduction

Tabular foundation models (TFMs) have recently emerged as a powerful paradigm for learning from heterogeneous tabular data, achieving state-of-the-art predictive performance by estimating the posterior predictive distribution of a label given all observed features (Müller et al., 2022; Hollmann et al., 2023). Despite their empirical success, existing TFMs rely purely on statistical associations, hence exploiting correlations that may not be stable under interventions, missingness, or distribution shift. In particular, they do not explicitly incorporate any notion of causal directionality (Pearl, 2009) between features. As a result, predictions may degrade sharply when mechanisms that are non-causally related to a given target variable change.

We illustrate this in Figure 1 with a chain $X \to Y \to Z$,

---

[*]Equal contribution [1]CISPA Helmholtz Center for Information Security, Saarbrücken, Germany. Correspondence to: Sascha Xu <sascha.xu@cispa.de>.

*Proceedings of the $2^{nd}$ ICML Workshop on Foundation Models for Structured Data*, Seoul, South Korea. 2026. Copyright 2026 by the author(s).

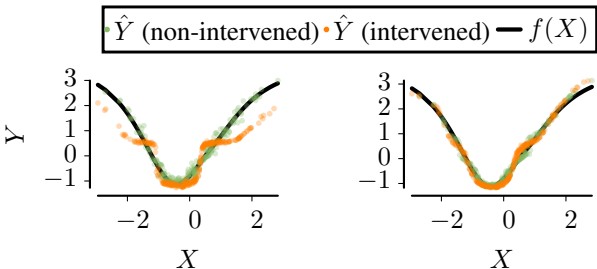

*Figure 1.* Prediction of a mediator $Y$ in a chain $X \to Y \to Z$. We measure test error without resp. with intervention on $Y \to Z$. TABPFN accurately models $X \to Y$ when no intervention is present (green), but fails under intervention (orange). In contrast, TABORDER remains accurate in both settings by first learning and then leveraging the causal order $X \to Y \to Z$.

predicting $Y$ from $X$ and $Z$. A state-of-the-art TFM such as TABPFN (Hollmann et al., 2025) leverages the statistical association between $Y$ and both of $X$ and $Z$. Although conditioning on the downstream $Z$ may achieve an error variance even below $\mathrm{Var}(N_Y)$ in an observational regime, this association may not be robust to changes in the data-generating process. When we intervene on the mechanism generating $Z$, for instance by removing the dependence between $Y$ and $Z$, the predictive performance of TABPFN degrades substantially (Fig. 1a). In contrast, a model that respects the causal structure and conditions only on the true parents of $Y$ remains robust (Fig. 1b).

Current TFMs do not attempt to model such structure, but focus on a supervised and label-centric training objective, that is, on learning $p(y \mid x, \mathcal{D})$ for a given target column. By contrast, our aim is to model the joint distribution of all variables, allowing prediction of any tabular cell from the remaining observed values. Since conditioning on all available features at all times is neither principled nor robust, we aim to learn how information should flow between variables in a way that reflects an underlying causal structure. A natural way to impose such structure is through an ordered factorization of the form $p(\mathbf{x} \mid \mathcal{D}, \pi) = \prod_i p(x_i \mid x_{\pi(1:i-1)}, \mathcal{D})$, where $\pi(i)$ denotes the $i$-th variable in a causal order.

Given these motivations, we introduce a tabular in-context learner, TABORDER, built to learn structured factorizations based on causal orderings $\pi$ inferred from data.

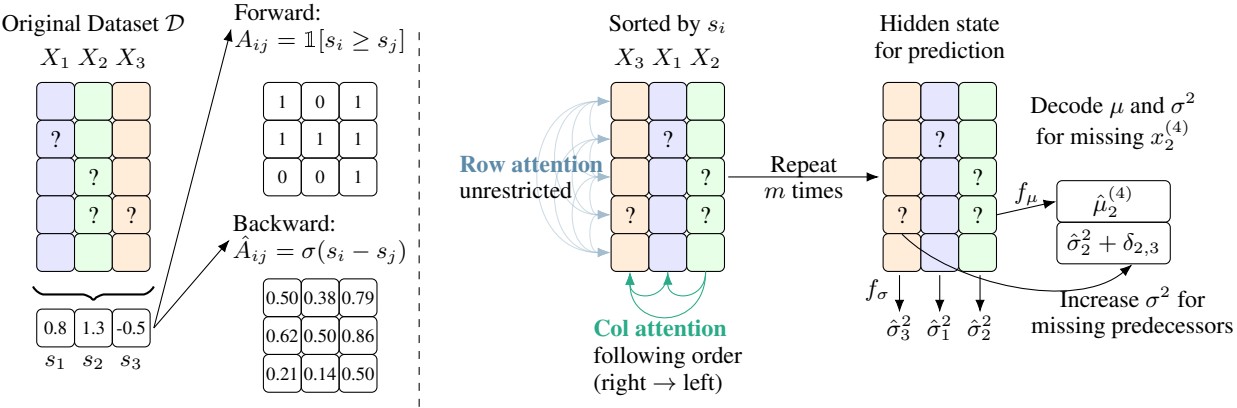

*Figure 2.* TABORDER Overview: We map each column of a dataset $\mathcal{D}$ to an order score $s_i$, from which we construct a hard/soft attention mask to constrain information flow (left). To predict missing entries, we alternate row-wise attention (unrestricted) and column-wise causal attention as per the order. Lastly, we decode the conditional mean and variance, accounting for missing potential causes (right).

## 2. Preliminaries

Let $\mathbf{X} = (X_1, \ldots, X_d)$ denote a collection of $d$ real-valued random variables drawn from an unknown joint distribution $P(\mathbf{X})$. We observe a dataset $\mathcal{D} = \{\mathbf{x}^{(r)}\}_{r=1}^n$, where each row $\mathbf{x}^{(r)} = (x_1^{(r)}, \ldots, x_d^{(r)}) \in \mathbb{R}^d$ is an independent realization of $\mathbf{X}$, and where some entries $x_i^{(r)}$ are missing, indicated by $m_i^{(r)} \in \{0, 1\}$.

Our goal is to learn structured conditional dependencies between the variables in $\mathbf{X}$. We base our approach off of **tabular foundation models** (TFMs) (Hollmann et al., 2023), which model the posterior predictive distribution of a target variable $x_i^{(r)}$ in row $r$ as $p_\theta(x_i \mid \mathbf{x}_{i*}^{(r)}, \mathcal{D})$, where $\mathbf{x}_{i*}^{(r)}$ denotes all entries in row $r$ except $x_i^{(r)}$. These models use all other variables as predictors, without accounting for potential causal directionality. To impose structure, we model the joint distribution of a row $\mathbf{x}^{(r)}$ via an **ordered factorization** with respect to an order $\pi \in S_d$. The value $\pi(k)$ denotes the index of the variable at position $k$ in the order. To predict an unknown entry $x_{\pi(k)}^{(r)}$, we condition only on preceeding columns as per

$$p_\theta(\mathbf{x}^{(r)} \mid \mathcal{D}, \pi) = \prod_{k=1}^d p_\theta\left(x_{\pi(k)}^{(r)} \mid x_{\pi(1:k-1)}^{(r)}, \mathcal{D}\right) . \quad (1)$$

## 3. TABORDER

Given a table $\mathcal{D} \in \mathbb{R}^{n \times d}$ with missing entries as per $M \in \{0, 1\}^{n \times d}$, TABORDER is a transformer-based architecture over individual cells $x_i^{(r)}$. Each cell $x_i^{(r)}$ and its missingness indicator $m_i^{(r)}$ are embedded through a learned linear map

$$\mathbf{e}_i^{(r)} = W_{\text{emb}} \begin{bmatrix} x_i^{(r)} \\ m_i^{(r)} \end{bmatrix} + \mathbf{b}_{\text{emb}} , \qquad W \in \mathbb{R}^{h \times 2} . \quad (2)$$

The cell embeddings are processed by alternating row-wise and column-wise self-attention layers, respectively

1. **without** causal masking, to induce an order over the given variable in the form of a score $s_i$, and

2. **with** a causal mask constructed from the learned scores, to model the conditional distribution of each cell.

### 3.1. Order Induction

To infer an ordering over variables, we apply a standard multi-head self-attention transformer (Vaswani et al., 2017) to the initial cell embeddings. We alternate row- and column-wise attention for $m$ steps that yield a hidden state $H^{\text{ord}} \in \mathbb{R}^{n \times d \times h}$. To obtain a column-wise score, we decode the learned embeddings of each cell using a two-layer MLP $f_{\text{ord}} : \mathbb{R}^h \to \mathbb{R}$ and average across rows to obtain

$$s_i(\mathcal{D}) = \frac{1}{n} \sum_{r=1}^n f_{\text{ord}}\left(H_{r,i}^{\text{ord}}\right) . \quad (3)$$

To impose the learned order $\hat{\pi}$ during attention in the predictive branch, we construct a causal mask. Column $i$ may attend to column $j$ only if $s_j \leq s_i$. To preserve differentiability, we employ a straight-through estimator: in the forward pass, attention is only permitted if

$$A_{i,j} = \mathbb{1}(s_j \leq s_i), \quad (4)$$

while in the backward pass we use a smoothed approximation of the former as per

$$\hat{A}_{i,j} = \text{sigmoid}\left(\frac{s_i - s_j}{t}\right) , \quad (5)$$

where the temperature $t$ controls the sharpness of the relaxation and is annealed during training.

## 3.2. Cell Prediction

Given the causal mask, we use a second transformer, again with alternating row- and column-wise attention, to learn structured conditional dependencies. Column-wise attention is restricted via the learned mask $A$, whereas row-wise attention remains unconstrained.

This produces predictive hidden states $H^{\text{pred}} \in \mathbb{R}^{n \times d \times h}$. We model the conditional distribution of each missing entry under the ordered factorization in Eq. (1) using **additive noise models**

$$X_i = f(X_{pa(i)}) + N_i, \ N_i \perp\!\!\!\perp X_{pa(i)} \ , \qquad (6)$$

where we model the noise to be Gaussian as per $N_i \sim \mathcal{N}(0, \sigma_i^2)$. We decode the pointwise conditional mean and the per-variable variance via two MLPs $f_\mu$ and $f_\sigma$ as

$$\hat{\mu}_i^{(r)} = f_\mu \left( H_{r,i}^{\text{pred}} \right), \quad \hat{\sigma}_i^2 = \frac{1}{n} \sum_{r=1}^{n} f_\sigma \left( H_{r,i}^{\text{pred}} \right) \ . \quad (7)$$

In addition, we also take into account the predictive uncertainty that arises for a cell $x_i^{(r)}$, if preceding values in that row $r$ are missing. We model this via a learnable variance increment $\delta_{i,j} \in \mathbb{R}^+$ decoded from the predictive hidden state, and thus obtain the final pointwise variance as

$$\hat{\sigma}_i^{2(r)} = \hat{\sigma}_i^2 + \sum_{j \in [d]} \delta_{i,j} \cdot \mathbb{1}(s_j < s_i) \cdot m_j^{(r)} \ . \quad (8)$$

The inclusion of this term increases the expressive power of TABORDER. Consider for example a collider structure $X_1 \rightarrow X_3 \leftarrow X_2$. When all variables are observed, the model predicts $X_3$ up to the noise variance. However, if either $X_1$ or $X_2$ is missing, the variance increment mechanism reflects the increased uncertainty due to the absence of causal information.

## 3.3. Training

All components of TABORDER are trained jointly by maximizing the log-likelihood of masked entries under the ordered conditional model. We assume access to an initially complete dataset with entries $x_i^{*(r)}$, from which we generate training instances by masking entries completely at random.

We visualize the flow of TABORDER in Figure 2. Given a masked dataset $(\mathcal{D}, M)$, we first infer an attention mask $A$ via the order induction module which is a standard tabular transformer. Conditioned on this mask, the predictive transformer produces cell-wise mean and variance estimates $\hat{\mu}_i^{(r)}$ and $\hat{\sigma}_i^{2(r)}$. The training objective is then to maximize the average log-likelihood of the masked entries,

$$\mathcal{L} = \frac{1}{|M|} \sum_{m_i^{(r)}=1} \log \left[ p_\mathcal{N}\!\left( x_i^{*(r)} \,\middle|\, \hat{\mu}_i^{(r)}, \hat{\sigma}_i^{2(r)} \right) \right] \ . \quad (9)$$

To maximize likelihood, the model must not only learn to infer missing values from context, but also to determine an order that aligns with the underlying causal structure. For certain structural causal models, such as **causal additive models**, (Bühlmann et al., 2014) show the maximum likelihood solution is only attained using a **topological order** of the underlying DAG.

Under these conditions, aligning the learned order with a topological order of the true DAG is necessary to attain the lowest possible loss, and thus the model is incentivized to learn a faithful causal order. However, as no general convergence guarantees are known for transformers, we empirically assess whether TABORDER recovers faithful causal orderings under various SCM generating mechanisms, and whether this improves predictions under intervention.

## 4. Related Work

**Tabular Foundation Models** are transformer-based in-context learners trained not on a single dataset but on large corpora of tables (Müller et al., 2022). TABPFN is a notable example that uses pre-training on millions of synthetic tasks drawn from a prior over structural causal mechanisms, achieving state-of-the-art performance without finetuning (Hollmann et al., 2023; 2025). There exist a variety of extensions, focusing on scalability (Qu et al., 2025) and retrieval on large real-world tables (Ma et al., 2025) for example.

Regarding causality, the focus so far has been on estimating treatment effects using TFMs (Robertson et al., 2025; Balazadeh et al., 2025). However, these approaches do not infer dataset-specific causal structure: they amortize over a synthetic SCM prior and do not estimate a causal graph or ordering for a new dataset. In general, causal structure learning has not been integrated into TFMs to date.

**Causal Discovery.** Our approach is conceptually related to methods for causal discovery that first estimate a causal order (Bühlmann et al., 2014; Rolland et al., 2022; Montagna et al., 2023a;b; Xu et al., 2025). However, all of these methods learn only the causal structure and perform neither prediction of missing values nor allow for in-context inference. AVICI (Lorch et al., 2022) is a notable exception that uses amortized inference to learn causal graphs across datasets. For training, it however requires supervision in the form of known graphs, potentially making it dependent on test-data priors (Montagna et al., 2025).

## 5. Experiments

**Causal Order Learning.** First, we evaluate the quality of causal orders that TABORDER recovers. To this end, we draw a ground-truth DAGs $G$ over a range of nodes $(5 - 10)$. Under a given $G$, we draw batches with sequence

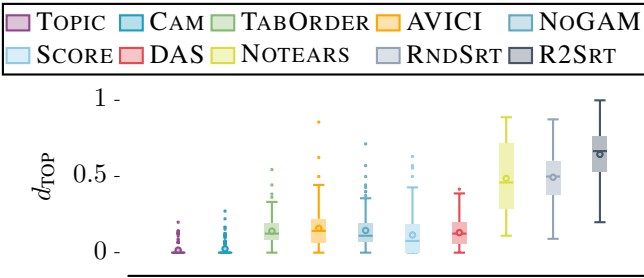

Figure 3. Average topological divergence ($d_{TOP}$, lower is better) between ground truth and estimated causal orders on synthetic datasets, generated from a nonlinear additive noise model with functions drawn from a Gaussian process.

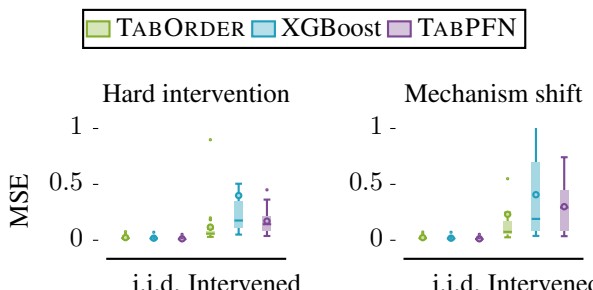

Figure 4. MSE for predicting $Y$ in the three-variable chain on an i.i.d. and an intervened sample $Y \rightarrow Z$. TABORDER remains accurate under both types of interventions by leveraging the learned causal order, while TABPFN and XGBOOST degrade significantly.

length $r \sim \text{Unif}_{\text{int}}(512, 1024)$ according to a non-linear SCM-generative process. We test both mechanisms that are additive, as well as non-additive across the inputs $pa_i$, here drawn from an RBF Gaussian process prior approximated by random Fourier features (RFFs). We describe the complete data-generating process in Appendix B.

We assess the causal order learning using the topological divergence $d_{TOP}$, which measures the discrepancy between the predicted causal order $\hat{\pi}$ and the ground-truth order implied by the true DAG $G^\star$ by counting edge disagreements

$$d_{TOP}(\hat{\pi}, G^\star) = \sum_{i=1}^{d} \sum_{j : \pi(i) \geq \pi(j)} \mathbb{I}[(i, j) \in G^\star] \,,$$

with lower values indicating better alignment with the true causal ordering. For comparison, we consider a selection of recent topological-ordering-based causal structure learning algorithms, CAM (Bühlmann et al., 2014), SCORE (Rolland et al., 2022), DAS (Montagna et al., 2023a), NOGAM (Montagna et al., 2023b), NOTEARS (Zheng et al., 2018) and TOPIC (Xu et al., 2025), as well as to the amortized learner AVICI (Lorch et al., 2022). To show the effect of simple heuristics, we include the baselines RNDSRT and R2SRT (Reisach et al., 2023).

We show the topological divergence $d_{TOP}$ across methods in Figure 3. TABORDER discovers causal orderings of matching quality compared to the outputs of structure learning algorithms tailored to this task. Methods based on regularized-likelihood scores under non-linear regressors, such as CAM, as well as the AMC-based TOPIC perform exceedingly well in our regime, which gives empirical support that the order can be accurately recovered by minimizing residual variance. These structure learning algorithms are however designed for one-shot learning given a single dataset. Albeit within a margin of error, TABORDER comes close to these approaches, while not relying on explicit regression or structural supervision as AVICI. Finally, we note that the performance of R2SRT indicates that the learning task on

this data is not amenable to simple sorting criteria. We include further experiments on both-additive and non-additive mechanisms in Appendix C, which TABORDER ably handles, and on scalability and ablations in Appendix E-F.

**Robustness under Intervention.** To conclude, we revisit the interventional setting introduced in Figure 1 in the Introduction. We consider a three-variable SCM $X \rightarrow Y \rightarrow Z$ with non-linear mechanisms, where we want to predict the mediator $Y$ from observations of $X$ and $Z$. We compare TABORDER against TABPFN and XGBOOST. On the test set, we remove the dependency to $Z$ entirely or change the mechanism for $50\%$ of the samples. Fig. 4 summarizes the results. All methods work well on data points drawn from the same distribution. However, when the mechanism for $Z$ is changed, TABPFN and XGBOOST suffer from a significant error increase as they rely on the now spurious correlation between $Z$ and $Y$. TABORDER, on the other hand, remains robust due to its stringent causal constraints, though it can incur a large error when the order is misidentified. Overall, the results demonstrate that TABORDER's causal-order-aware architecture provides robustness under interventions, highlighting the benefits of learning and leveraging causal structure in TFMs.

## 6. Conclusion

In this work, we introduced a tabular foundation model, TABORDER, to both learn and enforce causal orderings among tabular columns. Our empirical evidence suggests that transformer-based learning of causal structures is indeed effective without explicit supervision and helps preserve predictive performance under distribution shifts.

**Limitations and Future Work.** TABORDER currently supports only real-valued features, and our theoretical guarantees apply primarily to settings such as additive noise models. Extending the approach to mixed-type data, and establishing corresponding theoretical properties, presents an important direction for future work.

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

## A. Model and Training Details

This section summarizes the model components and hyperparameters used in the experiments. We implement the attention mask as an additive bias to the attention scores before applying the softmax. Forward, we use $A_{ij} \cdot \beta$, where $\beta$ is annealed from $-5.0$ to $-20.0$ during training; backward, we the sigmoid form $\hat{A}_{ij} = \sigma((s_i - s_j)/t)$ where $t$ is annealed from $1.0$ to $0.1$.

- We embed every cell to a vector of dimension $d_{\text{embedding}} = 128$.

- Transformer blocks: $6$ repetitions for inducing order and $4$ repetitions for cell prediction. We use Pytorch's `nn.TransformerEncoderLayer` with GELU activations, feed-forward dimension of $2 \cdot d_{\text{embedding}}$, dropout $0.1$ and prenorm.

- Optimization: Adam with base learning rate $2 \times 10^{-4}$, warmup ratio $0.03$ and weight decay.

- Batch size $4$ with 25000 training steps for a total of 100,000 datasets seen. Datasets are generated with $5 - 10$ variables each and $512 - 1024$ samples.

- Masking during training: random masking with `masking_percentage`=0.2.

## B. Synthetic Data Generation

For each synthetic example, we sample the number of variables $d \sim \text{Unif}_{\text{Int}}(5, 10)$, and construct a directed acyclic graph (DAG) $G = (V, E)$ over $V = \{1, \ldots, d\}$ by first sampling a random topological ordering and then sampling parent sets subject to an in-degree constraint $|pa_i| \in \{1, 2, 3\}$,.

If $pa_i = \varnothing$, the generator samples $X_i$ from a simple mixture of base distributions, with probability $1/2$, a scaled Gaussian $X_i = sZ$ where $Z \sim \mathcal{N}(0, 1)$ and $s \sim \text{Unif}(0.5, 2.0)$; otherwise, a scaled uniform $X_i = (U - \frac{1}{2})a$ where $U \sim \text{Unif}(0, 1)$ and $a \sim \text{Unif}(1.0, 5.0)$. Values are lightly clipped to avoid extreme outliers.

In the non-additive setting, each structural function $f_i$ is a nonlinear function of all parents jointly, sampled from an RBF Gaussian-process prior approximated by random Fourier features (RFF). Let $x \in \mathbb{R}^P$ denote the parent vector (after per-parent standardization), where $P = |pa_i|$. We sample a lengthscale

$$\ell \sim \exp\big(\text{Unif}(\log 0.3, \log 1.0)\big),$$

and define

$$\phi(x) = \sqrt{\frac{2}{H}} \cos\big(W^\top x + b\big), \qquad W \in \mathbb{R}^{P \times H}, \; b \in \mathbb{R}^H,$$

with $H = 256$ features, $W_{jk} \sim \mathcal{N}(0, \ell^{-2})$, and $b_k \sim \text{Unif}(0, 2\pi)$. Finally, with weights $a \sim \mathcal{N}(0, I_H)$,

$$f_i(x) = a^\top \phi(x).$$

This construction yields a smooth nonlinear mechanism that is explicitly non-additive across parents.

## C. Causal Order Learning

Our baselines for causal order discovery include topological sorting methods using regressors together with regularized likelihood or AMC-based scores, topological-ordering-based causal structure learning algorithms, CAM (Bühlmann et al., 2014), SCORE (Rolland et al., 2022), DAS (Montagna et al., 2023a), NOGAM (Montagna et al., 2023b), NOTEARS (Zheng et al., 2018) and TOPIC (Xu et al., 2025). We use their publicly available implementations, among others included in the `causal-learn` resp. `do-discover` Python packages, where we use default parameter choices suggested in the original implementations. A suite of methods, such as the classical PC algorithm (Spirtes et al., 2001), return only partially oriented structures (CPDAGs or PDAGs) that do not correspond to a unique causal order, and given that they are often outperformed by the methods we consider here in terms of other structural metrics (e.g., Xu et al., 2025), hence we omit them from comparison. In the evaluation, we use batch sizes of $|B| = 4$ for $n_B = 30$ batches, resulting in 120 runs per method.

We plot the topological divergence in Figure 5, including both the non-additive mechanisms shown in the main text as well as additive mechanisms, with the additional baseline R2SORT shown.

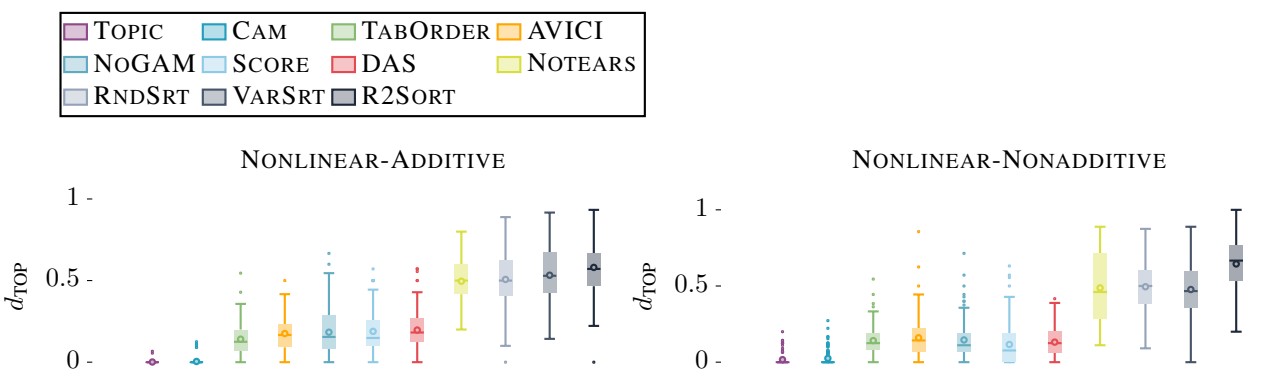

*Figure 5.* Average topological divergence ($d_{\text{TOP}}$, lower is better) between ground truth and estimated causal orders on synthetic datasets, generated from a nonlinear additive noise model with functions drawn from a Gaussian process, with both a variant where functional mechanisms are additive, respectively non-additive, in the causal parents.

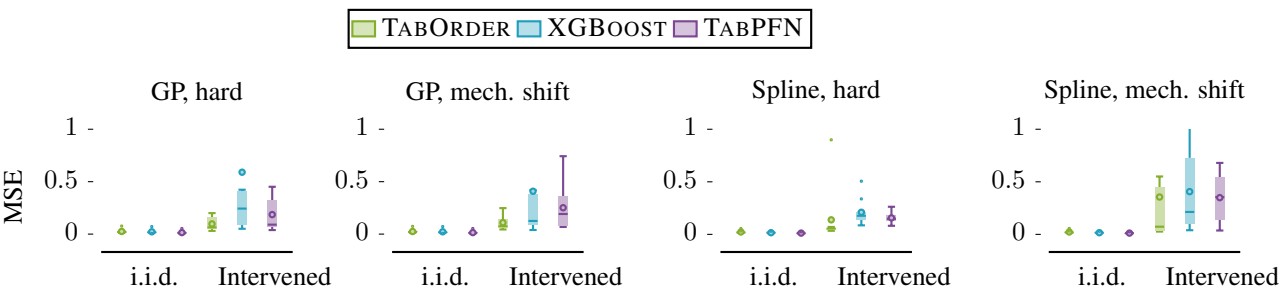

*Figure 6.* Test MSE for predicting $Y$ in the three-variable chain, split by mechanism family (GP vs. spline) and intervention type.

## D. Intervention Experiments

We generate data from a three-variable chain $X \to Y \to Z$. $X$ is sampled from a standard normal distribution. $Y$ is generated from an additive noise model $Y = f(X) + N_Y$ where $f$ is a GP sample with RBF kernel or spline function and $N_Y \sim \mathcal{N}(0, 0.1^2)$. $Z$ is generated similarly as $Z = g(Y) + N_Z$ where $g$ is an independent GP/spline and $N_Z \sim \mathcal{N}(0, 0.1^2)$. We generate training datasets with 5000 samples and test datasets with 2500 samples. We use the default configuration of TABPFN obtained via huggingface as well as the defualt XGBoost regressor with 100 estimators. We evaluate against TABORDER which was trained as detailed in Appendix A.

## E. Scalability Experiments

We also demonstrate the scalability of TABORDER depending on (1) the number of features $d$ and (2) sequence length of each batch $n$, otherwise using the standard data generating settings in Section B. For demonstration, we evaluate the main pretrained TABORDER model (additive GP-based) using checkpoints trained on 5–10 variables. For each configuration, we generate 25 batches of size 4 (a total of 100 graphs) and report averaged metrics. We vary either the number of variables $d \in \{10, 20, 50, 100, 200\}$ at fixed sequence length $n = 1024$, or the sequence length $n \in \{128, 256, 512, 1024, 2048\}$ at fixed $d = 10$. All experiments were run on a single `NVIDIA A100` GPU (40GB VRAM) using `PyTorch` without distributed computation.

Figure 7 shows the results, where we report the topological divergence (top. div.) and record runtime (evaluation wall-clock time) and peak GPU memory usage for each batch.

Computationally, both runtime and memory scale predictably with problem size, where inference time of learned orders is near-instant once the forward pass is completed. For increasing $n$ the scaling in evaluation time and memory usage is less steep compared to $d$. Order quality degrades moderately as the number of features $d$ increases, as predicting a consistent global ordering becomes more challenging in higher dimensions, however the degradation is gradual and remains below random baselines. In contrast, the topological divergence remains relatively stable across all considered sequence lengths

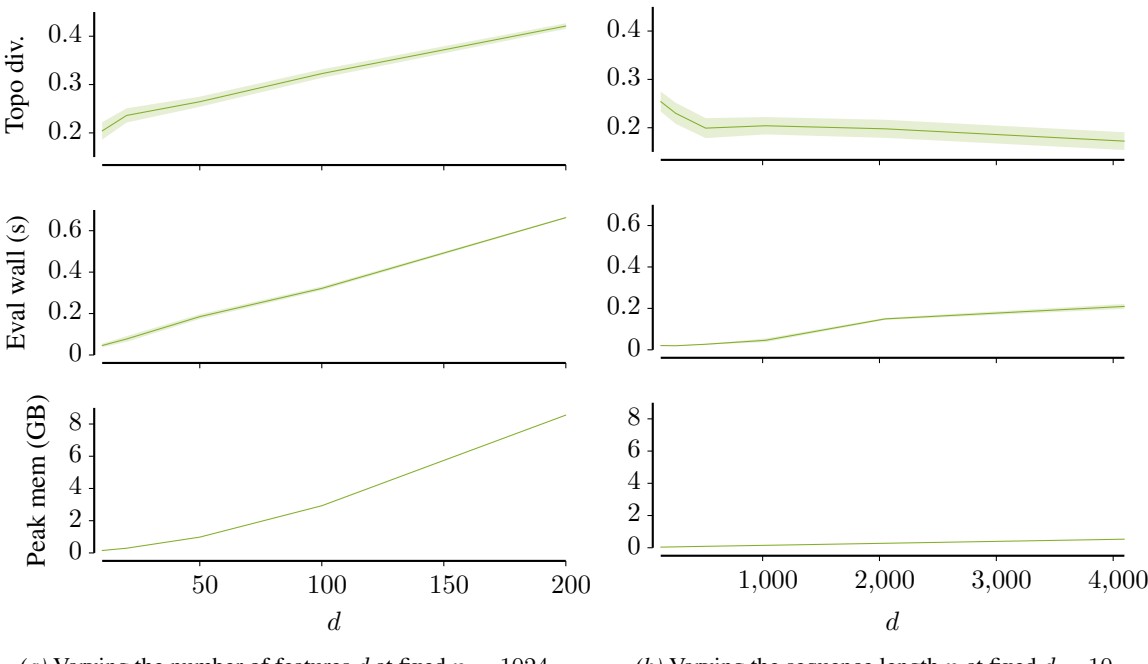

*(a)* Varying the number of features $d$ at fixed $n = 1024$.  *(b)* Varying the sequence length $n$ at fixed $d = 10$.

*Figure 7.* Scalability of TABORDER (model trained on additive GPs) by the number of features $d$ (left) and sequence length $n$ (right). Shaded regions show 95% confidence intervals.

with a slight improvement for larger $n$. This suggests that the model is not strongly data-limited in this regime and already extracts sufficient signal from relatively short sequences.

## F. Ablation Experiments

In Figure 8 we also investigate the effects of the training prior and supervision style during training (horizontal) depending on the functional form that we evaluate on (vertical). Specifically, we train TABORDER on functional forms (spline, Gaussian process, or a mix of functional forms) and with or without including additional loss supervision, i.e., providing true causal orders during training (where * denotes supervised variants).

As Figure 8 shows, a mix of generating processes (mixed) allows the model to generally fit all data-generating processes well compared to specialized (spline, GP) models. While training with explicit supervision (*) further reduces $d_{\text{TOP}}$, this may be prone to overfitting the synthetic data generators and, as shown in Figures 5, is not necessary to achieve competitive performance with the causal discovery baselines, hence we report the main variant (without *) as our main method.

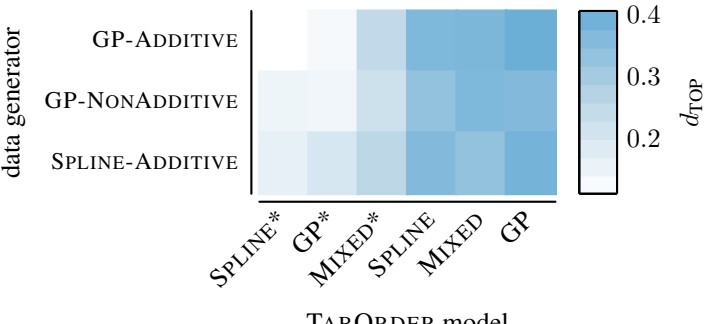

*Figure 8.* Shown are the effects of training TABORDER on specific functional forms (spline, Gaussian process, or a mix of functional forms), and of providing it with the true causal orders during training (denoted *) when evaluated on different synthetic data generators.

