# OpenReview forum: "Inducing Causal Order through Tabular In-Context Learning"
_ICML.cc/2026/Workshop/FMSD — FMSD @ ICML 2026 Poster_

### Official Review · Reviewer_nF8u · 2026-05-21
**Promising causal-order-aware TFM, but robustness evidence remains narrow**

**Rating:** 6
**Confidence:** 3

**Review:**

### Summary

The paper proposes TABORDER, a tabular in-context model that learns an ordering over columns and uses this order to restrict attention.

The goal is to avoid using downstream variables that may be predictive in observational data but unreliable under interventions.

The paper shows that TABORDER can recover useful causal orders on synthetic SCMs using $d_{\mathrm{TOP}}$, and tests robustness on a simple $X \to Y \to Z$ chain under hard intervention and mechanism shift.

### Strengths

The problem is important. Current tabular foundation models can use all correlations in the table, but not all correlations are stable when the data-generating process changes.

The architectural idea is simple and interesting. Learning a column order and using it as an attention constraint is a clean way to add causal directionality into a TFM.

The causal-order recovery results are useful. The paper compares TABORDER with several causal discovery baselines, and the results suggest that the learned order is not trivial.

The intervention example explains the motivation well. TabPFN and XGBoost use the association between $Y$ and $Z$, but this becomes harmful when the downstream mechanism changes.

### Areas for Improvement

The main weakness is that the robustness evidence is narrow. The intervention setting is still a three-variable chain. This shows the intended failure case, but it is not enough to support a broad robustness claim.

The paper does not show the cost of the causal mask on normal tabular prediction. A comparison with TabPFN on standard observational datasets would help.

The assumptions behind causal-order recovery need more discussion. Observational causal ordering is not identifiable in general, especially with hidden confounding or Markov-equivalent structures.

The method only handles real-valued features. This limits the practical TFM claim, since many tabular datasets contain categorical and mixed-type variables.

### Questions / Suggestions

How much standard predictive accuracy is lost by using the causal mask?

Can the intervention experiment be extended to larger SCMs with more variables?

What happens when the learned order is partly wrong?

Is the variance-increment term necessary, or are most gains coming from the order mask?

The paper should state more clearly under what assumptions likelihood training can recover a valid causal order.

### Justification of Score

I lean marginal accept. The paper has a good idea and the order-recovery results are promising. However, the strongest claim about robustness is still supported by a small intervention setup. I see this as a good workshop paper and a useful proof of concept, but not yet a fully validated TFM approach.

---

### Official Review · Reviewer_3bJB · 2026-05-21
**The paper introduce a tabular in-context learner, TABORDER, a transformer based architecture built to learn structured factorizations based on causal orderings inferred from data. The key idea is to learn a causal ordering over table columns and use this ordering to restrict transformer attention. The model learns a continuous score for each column to build an attention mask. This mask restricts the column-wise attention mechanism, ensuring information only flows from causal predecessors to downstream effects ($p(x) = \prod_k p(x_{\pi(k)} \mid x_{\pi(1:k-1)})$).**

**Rating:** 7
**Confidence:** 4

**Review:**

**Strengths**
- Learns causal ordering in tabular data.
- Cool and nice use of transformers to learn the causal ordering.
- Good preliminary experiments for the workshop.

**Areas of improvement**
- Related Works: InterpreTabNet [1] looks at Gumbel based hard masking for downstream predictions. Maybe a discussion against this + other related works that tackles masking  in tabular predictions would be nice too.
- Apply method to real-world datasets.

References
[1] Si, Jacob, et al. "InterpreTabNet: Distilling predictive signals from tabular data by salient feature interpretation." arXiv preprint arXiv:2406.00426 (2024).

---

### Official Review · Reviewer_1Yuk · 2026-05-22

**Rating:** 7
**Confidence:** 3

**Review:**

This paper proposes using causal order when predicting missing values in tabular tasks. The proposed method contains two transformer components. The first component is used to identify the causal order between variables and the second transformer is used to make predictions with an uncertainty estimate.

The problem statement is well-motivated. Authors also show the evidence that the problem exists. The proposed solution of learning a latent ordering and using it to constrain transformer attention is interesting.

However, the proposed architecture is trained and tested only on synthetic data. Despite comparing with multiple baselines, we cannot strongly conclude that the method would generalize to real structured data due to limited evaluation.